# Carriers and Antigens: New Developments in Glycoconjugate Vaccines

**DOI:** 10.3390/vaccines11020219

**Published:** 2023-01-19

**Authors:** Robert M.F. van der Put, Bernard Metz, Roland J. Pieters

**Affiliations:** 1Intravacc, P.O. Box 450, 3720 AL Bilthoven, The Netherlands; 2Department of Chemical Biology & Drug Discovery, Utrecht Institute for Pharmaceutical Sciences, Utrecht University, P.O. Box 80082, 3508 TB Utrecht, The Netherlands

**Keywords:** antimicrobial resistance, carbohydrates, carrier proteins, ESKAPE pathogens, glycoconjugate vaccines, glycoengineered proteins, outer membrane vesicles, protein nanocages, virus-like particles

## Abstract

Glycoconjugate vaccines have proven their worth in the protection and prevention of infectious diseases. The introduction of the *Haemophilus influenzae* type b vaccine is the prime example, followed by other glycoconjugate vaccines. Glycoconjugate vaccines consist of two components: the carrier protein and the carbohydrate antigen. Current carrier proteins are tetanus toxoid, diphtheria toxoid, CRM197, Haemophilus protein D and the outer membrane protein complex of serogroup B meningococcus. Carbohydrate antigens have been produced mainly by extraction and purification from the original host. However, current efforts show great advances in the development of synthetically produced oligosaccharides and bioconjugation. This review evaluates the advances of glycoconjugate vaccines in the last five years. We focus on developments regarding both new carriers and antigens. Innovative developments regarding carriers are outer membrane vesicles, glycoengineered proteins, new carrier proteins, virus-like particles, protein nanocages and peptides. With regard to conjugated antigens, we describe recent developments in the field of antimicrobial resistance (AMR) and ESKAPE pathogens.

## 1. Introduction

Vaccines have proven to be one of the most effective applications to prevent and protect against infectious diseases. As such, society relies heavily on existing vaccines, but also on the development of new vaccines and vaccine platforms. It has already been more than five centuries since the first strategies to battle infectious diseases were implemented [1]. However, it was not until the late 18th century that Jenner came up with his strategy to battle smallpox, utilizing live-attenuated viruses [1]. In the 19th century, Pasteur developed a successor in the form of a live-attenuated vaccine protecting against rabies. New and more efficient production platforms were established when viruses, such as influenza, mumps, measles, rubella, and polio, were propagated on cells and bacteria, such as *Clostridium tetani* and *Corynebacterium diphtheriae,* were isolated and grown in a culture medium for the production of toxoid vaccines (tetanus and diphtheria). The very first concepts for glycoconjugates were already evaluated in the early 1930s by Avery and Goebel [2]. However, the first conjugate vaccine targeting *Haemophilus influenzae* type b (Hib) was developed around 1985 [3,4], based on extracted polysaccharides (PS). Glycoconjugate vaccines appeared to be an important vaccine technology platform. Utilizing the PS is important since they aid bacteria in their ability to infect individuals and to evade the immune system and trigger an inflammatory response. Glycoconjugate vaccines consist of the extracted and purified PS alone or covalent attachment (conjugate) of these PS to carrier proteins. Vaccines containing purified PS alone do work, but they can only stimulate B-cells to produce antibodies and no immunological memory response. There are certain benefits in using a carrier protein with the PS conjugated to a carrier. Conjugation to a carrier protein significantly improves the immunogenicity towards a T-cell-dependent response to the PS [5,6]. This has been shown for PS originating from the capsular structures of Hib, *Streptococcus pneumoniae*, *Neisseria meningitidis* and *Salmonella* Typhi [7,8]. For children under the age of two, these conjugate vaccines are extremely effective, which is fortunate since the disease burden of encapsulated bacteria at that age is high [9,10]. Long-lasting immunity is induced by the conjugation to a carrier protein, triggering the immune system towards a T-cell-dependent immune response [11].

The use of conjugate vaccines has had some undesirable effects. Pichichero (2013) [12] and Broker (2017) [13] describe the interference of immunogenicity thoroughly and refer to two underlying mechanisms: antigen competition and carrier-induced epitope suppression (CIES). Antigen competition is caused by the processing of antigens in combination vaccines. On these specific occasions, a protein (e.g., tetanus toxoid) is co-administered as an independent vaccine constituent (against tetanus) in combination with a conjugate vaccine using the same protein, but now as a carrier (e.g., DaKTP-Hib-HepB). CIES is attributed to a pre-existing vaccination using the carrier protein as the single vaccine component as well as the carrier for the conjugate vaccine or simply co-administration. It became apparent that there was a diminished immune response observed towards the conjugated polysaccharide antigen [14,15,16,17]. CIES is difficult to predict for any particular glycoconjugate vaccine. Based on immunological data from pre-clinical animal studies, it is still difficult to estimate the outcome of a clinical study. The introduction of new and different carriers originating from the same species as the immunogen could potentially circumvent poor immunogenicity. It could also induce a broader application for targeting multiple serotypes [18] and would at least have the potential to partially solve these issues attributed to antigen competition and CIES. The selection and development of carriers and PS have led to exciting combinations and innovative approaches. In this review article, we provide an overview of recent innovations in the development of novel conjugate vaccines. The vaccine carriers include new proteins, virus-like particles (VLP), (glycoengineered) Outer Membrane Vesicles (OMVs) and proteins nanocages and peptides. Furthermore, this review discusses the latest development of the antigenic parts of a conjugate vaccine, aimed to protect against AMR-designated ESKAPE pathogens.

## 2. Conjugate Vaccine Carriers

The choice of a carrier for conjugate vaccines has to take certain criteria into account, of which a proven track and safety record is the most important. The first traditional carrier proteins utilized were, in fact, toxoids produced from tetanus and diphtheria toxins. Even though chemical detoxification of these toxins can lead to potential heterogeneity and unwanted modifications, they have proven to be safe and efficacious. A second selection criterium was that these carriers could be produced at a large scale under GMP conditions. Several other additional criteria cannot be disregarded. New carriers will also have to be evaluated for their immunogenicity after conjugation, reproducible manufacturing and sufficient surface exposure of reactive amino acids for conjugation, solubility, and stability. Extensive physicochemical and immunological characterization is important for the selection of a carrier. Moreover, immunological responses of new carrier proteins have to be evaluated.

This information will be assessed by regulatory agencies, such as WHO, EMA or FDA. Additionally, when the carrier is also used as an antigen itself, it is key to evaluate at which site conjugation is performed. Here, the conjugation of antigens to potential B-cell epitopes can interfere with or reduce the immune response towards the carrier antigen. With the previously described CIES, the selection and development of new carrier proteins beyond the traditional currently licensed carriers seems inevitable. In the next sections, we discuss the traditional carrier proteins for glycoconjugate vaccines followed by a detailed evaluation of new carrier proteins.

### 2.1. Traditional Protein Carriers

The currently used carrier proteins in licensed vaccines are tetanus toxoid (TT), diphtheria toxoid (DT), CRM197, Haemophilus protein D (PD) and the outer membrane protein complex of serogroup B meningococcus (OMPC). TT and DT are the first protein carriers used for Hib conjugate vaccines due to their safety profile. The Hib conjugate vaccines were administered as stand-alone vaccines. They are derived from the chemical detoxification of bacterial toxins. CRM197 is a non-toxic mutant due to the detoxification of diphtheria toxin by means of genetic mutation. A single glycine to glutamic amino acid substitution at position 52 resulted in a non-toxic mutant [19]. There are multiple licensed vaccines using CRM197, of which Hib, multivalent meningococcal and pneumococcal conjugate vaccines are prime examples [20,21]. PD (cell–surface protein) are applied in a multivalent pneumococcal vaccine [22,23] and in an OMPC-based vaccine for both Hib and pneumococcal vaccines [24,25].

### 2.2. Other Protein Carriers

The groups of Broker et al. (2017) [13] and Micoli et al. (2018) [26] have given an overview of investigated carrier proteins actively applied in the field including analytical characterization criteria. A prime example was the recombinant non-toxic form of *Pseudomonas aeruginosa* exotoxin A (rEPA) used as a carrier for *Escherichia coli* [27,28], *Shigella* O-antigens, *Staphylococcus aureus* type 5 and 8 capsular PS and the *S*. Typhi Vi antigen [29,30,31,32]. Other examples included the pneumococcal recombinant proteins spr96/2021 and spr1875 [33]; proteins originating from the extra-intestinal pathogenic *E. coli*-derived Upec-5211 and Orf3526 [33]; a recombinant protein comprising of filaments of promiscuous human CD4^+^ T-cell epitopes [34,35]; and the recombinant tetanus toxin HC fragment [36]. Furthermore, they discuss several examples of synthetic peptide-based carriers, including the best-known example, a 13 amino acids non-natural pan DR (PADRE) universal helper T-lymphocyte epitope [37,38,39,40,41,42,43]. Lastly, they provide data on the use of proteins with the dual role of both carrier and antigen (detoxified pneumococcal-based pneumolysin [44], *S. aureus*-derived recombinant proteins [45], *Clostridioides difficile* toxin fragments [46], Group B Streptococcus pili proteins GBS80 and GBS67 [47,48] and flagellin as the carrier protein for *Salmonella enteritidis* O-antigen [49]) and the application of nanoparticles (KLH [50], Qβ virus-like particles [51] and glycoengineered outer membrane vesicles [52]).

### 2.3. Recent Developments Regarding Protein Carriers

In this section, we provide an overview of developments with respect to the application of different carrier proteins in the field of prophylactic vaccines in the last five years. A multitude of different proteins and other types of carriers have been investigated. Even though the traditional carrier proteins and other potential carriers (liposomes, polymers, inorganic gold particles, dendrimers and nanodiscs) are still very much applicable in the development of conjugate vaccines, here we have focused on six new and different types of carriers including: (1) Outer membrane vesicles (OMV) and generalized modules for membrane antigens (GMMA), (2) glycoengineered proteins and OMVs, (3) proteins, (4) virus-like particles (VLP), (5) protein nanocages and (6) peptides (Figure 1).

#### 2.3.1. Outer Membrane Vesicles/Generalized Modules for Membrane Antigens

The use of outer membrane vesicles (OMV) or generalized modules for membrane antigens (GMMA) provides a versatile and flexible platform for the development of monovalent or multivalent conjugate vaccines [53]. This platform is an excellent basis as a carrier and can be produced from a multitude of different bacteria [54]. The safety profiles of OMVs and GMMAs have been studied in several clinical studies [55,56,57]. Additionally, they are of particular interest for their self-adjuvant properties, governed by the presence of pathogen-associated molecular patterns (PAMPS), such as lipooligosaccharide (LOS) and lipoproteins [58,59,60]. Due to these characteristics, there is less need for the addition of adjuvants, e.g., aluminum hydroxide. In comparison to smaller carries, the sheer size of OMVs and GMMAs (50–200 nm) with their large surface area bearing a variety of properties, such as hydrophobicity, charge and the potential for receptor interaction, provide a significant constructive effect on the uptake by APCs [61]. Finally, a major benefit is that they can be efficiently manufactured using high-yielding production platforms [62,63]. Several groups provide ample data on the successful application of OMV, and GMMA as a carrier (Table 1). The group of Micoli et al. has reported extensively on their GMMA carriers originating from different pathogens and conjugating various polysaccharide and protein antigens [53,64]. Palmieri et al. have investigated conjugates using GMMAs originating from *S. enterica* and conjugating Group A *Streptococcus* cell wall oligosaccharide [65], whereas Scaria et al. describe the application of OMV, a membrane vesicle derived from *N. meningitidis* as the carrier for a malaria transmission-blocking antigen.

#### 2.3.2. Glycoengineered OMVs and Proteins

Where traditional glycoconjugate vaccines rely on the extracted or synthetically designed saccharides being chemically conjugated to a carrier protein, in vivo protein glycan coupling technology (PGCT) is achieved through recombinant enzymatic construction of a glycoconjugate directly in bacteria [69]. These glycoconjugates consist of a carrier protein and a carbohydrate moiety originating from the bacterial capsule or the O-antigen polysaccharide (O-PS) [70]. A prerequisite here is that the O-PS structure has been identified, including the gene cluster responsible for its production. PGCT is based on the *Campylobacter jejuni N*-linked glycosylation system, which can be expressed in *E. coli*. Moreover, the *E. coli* mutant is capable of producing heterologous polysaccharides on its glycosyl carrier lipid [71]. The final construction of the glycoconjugate requires the presence of genome clusters of the bacterial polysaccharide, which is achieved by a plasmid encoding for the carrier protein, including oligosaccharyl transferase (OTase; characteristically originating from PglB from *C. jejuni* [71]. Bioconjugation in general leads to heterogeneously expressed glycoproteins. The polysaccharides themselves show polydispersity, and the number of carbohydrate repeat units can vary significantly. Additionally, the conjugation rate, or quantity of polysaccharides conjugated to one single carrier protein, varies as well. For these reasons, the structure elucidation of the conjugate is not evident and requires several methods [72,73]. Next to this, there are some other challenges, such as OTases, that are not fully compatible with most bacterial O-SP, resulting in even more heterogeneous bioconjugates. Nevertheless, PGCT does not need intermediate steps, such as glycan synthesis, purification, conjugation, and process controls. As such, PGCT is potentially more cost-effective. This new and exciting field of conjugate vaccines is already well studied (Table 2). Nicolardi et al. describe the development of a protein-based glycoengineered vaccine using the exotoxin A from *P. aeruginosa* as a carrier targeting *E. coli* serotypes O2, O6A and O25B [72]. Harding et al. report (1) on a polyvalent pneumococcal bioconjugate vaccine using the natural acceptor protein ComP as a vaccine carrier decorated with *S. pneumoniae* CPSs [74] and (2) *K. pneumoniae* K1 and K2 serotypes produced in glycoengineered *E. coli*.

#### 2.3.3. Protein-Based Carriers

Antigen competition and CIES pose a problem for the inclusion of antigens with respect to their use of vaccines. For this reason, the introduction of new protein carriers that improve upon the traditional carriers is much needed. Recent developments in conjugate vaccine design include proteins from both homologues and heterologous origin as carriers of conjugated antigens. From this, either a monovalent or bivalent (combining two target antigens) approach is used to target specific pathogens (Table 3). Wang et al. targeted *S. pyogenes* (Group A *Streptococcus*, GAS) cell wall oligosaccharides (GAC) conjugated to the streptococcal C5a peptidase carrier as a bivalent conjugate vaccine [82,83]. Furthermore, Kapoor et al. also target the GAC oligosaccharides from GAS. However, they found an application in the use of the secreted toxin antigen streptolysin O (SLO) as the protein carrier [84]. Additionally, Romero-Saavedra et al. described the use of two enterococcal proteins (secreted antigen A and the peptidyl-prolyl cis-trans isomerase) as a carrier for the *Enterococcus faecalis* polysaccharide di-heteroglycan [85].

#### 2.3.4. Virus-Like Particles

Virus-like particles (VLPs) are part of the family of subunit vaccines. VLPs are formed by means of viral capsid protein self-assembly. The formed particles mimic the original virus but cannot replicate or induce an infection. This is an advantage with respect to the risks associated with live-attenuated vaccines [98]. Next to serving as a vaccine component themselves, VLPs also provide a platform as a carrier. Similar to chemical conjugation, glycoengineered proteins and OMVs, protein display can be used for VLPs to present heterologous protein antigens (Table 4). An example is reported by Basu et al., who in one study, conjugated two Zika virus proteins (MS2 and PP7) and, in another study, coupled multiple synthetic peptides mimicking Chikungunya virus B-cell epitopes to a canonical ssRNA phage Qβ VLP (Qβ) [99,100]. Warner et al. used the same strategy, but conjugated synthetic peptides originating from the Dengue virus to the Qβ VLP carrier [101]. More recently, Zha et al. demonstrated that a recombinantly expressed SARS-CoV-2 receptor-binding domain (RBD) can be conjugated to VLPs, derived from the cucumber mosaic virus [102]. Carbral-Miranda et al. used the cucumber mosaic virus VLP carrier to successfully conjugate a Zika virus E-DIII protein [103].

#### 2.3.5. Protein Nanocages

A comprehensive overview of the application of nanocages is given by Curley et al. [109]. Nanocages are constructed from non-viral protein subunits and have similarities to VLPs. A number of these subunits can self-assemble into nanocages. They form symmetrical structures that differ from VLPs in terms of shape and size [110]. Nanocages contain repetitive structures recognized by B-cell receptors. One specific example where nanocages are utilized as a carrier for vaccine development is ferritin-based nanocages. These ferritin modules derived from *Helicobacter pylori* assemble into a 12 nm diameter spherical cage and are composed of 24 subunits with a hollow core [111]. An interesting aspect here is the application of the innovative SpyCatcher-SpyTag conjugation approach as described by Chen et al. [112] and Wang et al. [113]. This method of protein ligation is based on the modified domain of the *S. pyogenes* surface protein (SpyCatcher) that specifically recognizes a 13-amino-acid peptide (SpyTag). A covalent isopeptide bond is formed between the side chain of lysine in the SpyCatcher and aspartate in the SpyTag. This results in covalently linked protein complexes [114]. Several vaccine targets have been investigated that use protein carrier nanocages (Table 5), including hemagglutinin from the influenza virus [115] and the preS1 protein from hepatitis B [113].

#### 2.3.6. Peptides

Peptide-based vaccines are a type of subunit vaccine predominantly comprised of short (single-epitope) sequences that can be produced synthetically. These types of vaccines have several advantages, such as their good safety profile, fast and simple manufacturability at a large-scale and high purity [117]. Additionally, designing peptide vaccines combining multiple single epitopes and creating synthetic long peptides into recombinant overlapping peptide vaccines could broaden their applications and effectiveness [118,119]. In most cases, the antigenic peptide has to be equipped with an adjuvant, e.g., the universal T-cell epitope PADRE, and also with polyleucine as a hydrophobic solubilizing moiety in order for them to be useable. Several exemplary applications are described in Table 6.

## 3. Conjugate Vaccine Antigens

Since the introduction of the first carbohydrate conjugate vaccine targeting *H. influenzae* type b [4], conjugate vaccines have been developed for many different applications in the prophylactic domain to prevent infectious diseases, such as pneumonia, meningitis and sepsis. Currently, there are close to twenty conjugate vaccines on the market, licensed by the FDA, EMA and WHO. These vaccines are all based on the chemical conjugation of capsular polysaccharides targeted against Hib, *N. meningitidis* serogroup ACWY, *S. pneumoniae* or *S*. Typhi. Recent developments show conjugate vaccines targeting other pathogens such as a multivalent Extraintestinal pathogenic *E. coli* bioconjugate (ExPEC), *Klebsiella* O-antigen based, Group B *Streptococcus*, other serotypes of *S. pneumoniae*, PNAG (*N. meningitidis*, *S. aureus*, *Neisseria gonorrhea*, *Klebsiella pneumoniae*, *E. coli* and *S. pneumoniae*), *Shigella flexneri* 2a and others [124,125].

Next to the area of infectious diseases, carbohydrate-based conjugate vaccines find their application in the therapeutic domain targeting different forms of cancer. Often these vaccines are based on synthetic moieties mimicking tumor-associated carbohydrate antigens (TACAs) [126] including protein-linked Tn, Sialyl-Tn (STn) and TF (conjugated to the -OH group of serine) and glycolipid-based gangliosides GM2, GD2, GD3, fucosyl-GM1, Globo-H and Lewis^y^. The advantages of vaccination and targeting these TACAs include the significant reduction in toxicity and invasiveness compared to cytotoxic therapies such as radio- and chemotherapy [127]. Soriaul et al., Shivatare et al. and Hossain et al. gave an excellent overview of the current field of carbohydrate-based cancer vaccines. This included potential carrier proteins, adjuvants, and modifications of TACA antigens to improve stability, reduce hydrolytic cleavage and the cost of large-scale chemical or enzymatical synthesis [128,129,130]. Another area where carbohydrate-based conjugate vaccines find their application is the field of neurodegenerative diseases such as Alzheimer’s disease [131]. Overall, it cannot be denied that conjugate vaccines find their use in the broadest sense stretching multiple disease areas.

Despite the broad potential, the importance of the battle against antimicrobial resistance pathogens (AMR) cannot be overstated. Resistance to antibiotics can occur naturally in bacteria. However, the main reason for AMR is related to incompliance with respect to dosage and in adherence with the duration of the treatment. If antibiotic resistance would confine itself to a single pathogenic species or antibiotic, it would potentially still be manageable. At this moment, several bacteria show multidrug resistance (MDR), meaning they are resistant to multiple antibiotics. MDR is even more problematic specifically due to bacteria’s abilities to transfer AMR-related genes between different species (Table 7).

The so-called ESKAPE pathogens (*Enterococcus faecium*, *Staphylococcus aureus*, *Klebsiella pneumoniae*, *Acinetobacter baumannii*, *Pseudomonas aeruginosa* and *Enterobacter* spp.) have developed MDR against macrolides, lipopeptides, fluoroquinolones, oxazolidinones, β-lactams, tetracyclines, β-lactam–β-lactamase inhibitor combinations and against the last line of defense antibiotics such as glycopeptides, carbapenems and polymyxins [132]. Murray et al. (2019) estimated that the global burden of AMR included 23 pathogens and 88 pathogen–drug combinations spreading across 204 territories and countries. They attributed 1.27 million deaths directly to AMR and even a staggering 4.95 million deaths associated with AMR. This makes AMR the third largest global killer due to an infectious agent after COVID-19 and tuberculosis [133]. Now, looking past the COVID-19 pandemic, AMR can be classified as the new overlooked pandemic [134,135]. WHO and CDC have produced priority lists on which AMR targets have been classified (Figure 2 [136,137]).

A collaborative study performed in 2019 (European Antimicrobial Resistance Collaborators, EARC) provided confirmational data that six of the ESKAPE pathogens that are at the top of AMR list, are each responsible for more than 25.000 associated deaths in Europe in 2019 alone (Figure 3 [138]). While the list extends much further than the six ESKAPE pathogens alone, both agencies rate all the ESKAPE pathogens at the top of their priority list.

Glycoconjugate vaccines have proven to be successful in the battle against infectious diseases and potential AMR targets. Current developments show that substantial progress is made in the development of conjugate vaccines applying complex glycans obtained by either extraction, chemical synthesis or through bioconjugation. Here, we provide an overview of the individual AMR challenges for each of the ESKAPE pathogens and the recent applications of carbohydrate conjugate vaccines targeting AMR in the last 5 years.

### 3.1. Enterococcus faecium

*E. faecium* is a Gram-positive commensal bacterium that is associated with severe infections in immunocompromised populations. Infections are persistent in the hospital environment with common hospital-acquired infections (bloodstream and urinary tract infections) in patients suffering from other underlying conditions. Up to 30% of all healthcare-related enterococcal infections are resistant to vancomycin, with increasing resistance towards other antibiotics [136,137]. Primary targets for vaccine development include capsular polysaccharide, cell wall teichoic acid and lipoteichoic acid. Another polysaccharide which is anchored to the peptidoglycan layer with possible immunogenic properties is the enterococcal polysaccharide antigen [139].

Kalfopoulou et al. provided data on two highly specific antibodies directed against the polysaccharide and carrier protein. Both antibodies showed good opsonic killing against target bacterial strains in upwards of 40% for *E. faecium* and 90% for *E. faecalis* respectively [140]. Romero-Saavedra et al. showed similar results with antibodies against the corresponding conjugate, as well as against the respective protein and carbohydrate antigens. This included effective opsonophagocytic killing against different *E. faecalis* and *E. faecium* strains and recognition of 22 enterococcal strains [85]. Zhou et al. took a different approach with a synthetic cell wall teichoic acid (Figure 4), and also demonstrated a robust immune response and high antibody titers [141]. All three described examples provided successful data and are promising vaccine candidates against *E. faecium* targeting different antigens. They also included homologues carrier proteins, thus providing potential broader protection among multiple serotypes (Table 8).

### 3.2. Staphylococcus aureus

*S. aureus* is a commensal Gram-positive bacterium associated with methicillin resistance (MRSA) and not only hospital-acquired infections in patients suffering from other underlying conditions, but also as a communal acquired infection of the skin and soft tissue. MRSA results in increased morbidity, mortality, length of hospitalization of patients and incurred costs [136,137]. Vancomycin is the first line antibiotic in the treatment of MRSA infections [147]. However, both intermediate and complete resistance to vancomycin has developed among clinical isolates within the past decades [148]. While the incidence of vancomycin resistance is still rare, it has become a serious public health concern [149]. The capsular polysaccharide from *S. aureus* has 13 different serotypes identified from clinical isolates of which types 5 and 8 are the most abundant [150]. Furthermore, the broadly associated poly-(1→6)-N-acetyl-D-glucosamine exopolysaccharide (PNAG) is also a potential target for vaccine development.

Gening et al. describe the successful development of a PNAG glycoconjugate including passive and active protection against different pathogens. Their conjugate 5GlcNH_2_-TT is being produced under GMP conditions and has entered safety and effectiveness testing in humans and economically important animals [151]. Zhao et al. provide data on the successful development of a synthetic capsular polysaccharide type 8 trisaccharide (Figure 4) conjugated to CRM197. They report a robust immune response including monoclonal antibodies recognizing the *S. aureus* strain [142]. While the prior two groups employ more traditional conjugation using carriers and antigens, the group of Stevenson et al. provided data on an OMV bioconjugate. They first showed the successful decoration of the *E. coli* OMV with PNAG followed by the introduction of an *S. aureus* enzyme responsible for PNAG deacetylation. Antibodies that were generated by vaccinating with this candidate provided efficient in vitro killing of *S. aureus* and *Francisella tularensis* as well [79]. All three described groups provided results that reveal the potential for effective vaccines targeting *S. aureus* utilizing different platforms (Table 9).

### 3.3. Klebsiella pneumoniae

*K. pneumoniae* is a commensal Gram-negative bacterium. It is associated with persistent urinary tract, soft tissue, and ventilator-associated pneumonia infections. Furthermore, it affects the elderly and immunocompromised patients causing pneumonia and sepsis. Infections can also be found in communal settings and are widespread. Resistance is found for third-generation cephalosporins and carbapenems. This resistance is conferred through extended-spectrum beta-lactamase (ESBL) or Amp-C beta-lactamase production. Last resort options to treat ESBL-producing Enterobacteriaceae infections are limited and can include the use of intravenous (IV) carbapenem antibiotics [136,137]. *K. pneumoniae* polysaccharides offer a wide array of targets for vaccine development, of which the capsular polysaccharide-derived K-antigen (77 serotypes) and lipopolysaccharide-derived O-antigen are the most promising [152]. Early developments of a 24-valent vaccine (1990s) have been shown to be effective in humans, but further development has been stopped due to overly complicated manufacturing [153].

Lin et al. suggest that traditional depolymerization of the CPS results in the loss of immunogenicity. They circumvent this by first extracting the polysaccharides from *K. pneumoniae* before applying CPS depolymerases identified from phages to cleave K1 and K2 (Figure 4) CPS, resulting in intact structural units of oligosaccharides without modifications. Conjugate vaccines employing TT as a carrier and using both K1 and K2 oligosaccharides were successful in protecting mice in a challenging study post-vaccination [143]. Feldman et al. reported on the recombinant production of a bioconjugate vaccine produced in glycoengineered *E. coli* targeting serotypes K1 and K2. The bioconjugates proved to be immunogenic and provided protection in mice when subjected to a lethal infection using two hypervirulent *K. pneumoniae* strains [77]. Ravinder et al. designed synthetic saccharides mimicking the K2 oligosaccharide. After investigating different quantities in repeat units they concluded that there was an optimum at five repeat units of the saccharide. Their glycoconjugate vaccine equipped with the hepta-saccharide exhibited good bactericidal activity [154]. There are multiple strategies being investigated in the field to design new vaccines targeting *K. pneumoniae* that include traditionally extracted polysaccharides, synthetic strategies and bioconjugates. This will yield multiple vaccine candidates with great potential (Table 10).

### 3.4. Acinetobacter baumannii

*Acinetobacter baumannii* is a commensal Gram-negative bacterium with great capability to spread in hospital settings. A particular challenge lies in the fact that it is frequently found at healthcare facilities. It is the leading cause of ventilator-associated pneumonia, and bloodstream, wound and urinary tract infections while being inherently resistant to various classes of antibiotics and easily acquiring resistance. It shows high mortality rates for invasive infections, particularly in carbapenem-resistant strains. Some *Acinetobacter* strains have built resistance to practically all antibiotics. Therefore, they are designated at the top of all AMR priority lists. While overall carbapenem-resistant *Acinetobacter* rates have decreased, gene transfer between other species, making them carbapenemases competent, is an increasing problem [136,137]. *A. baumannii* shows great biodiversity in polysaccharide structures, of which more than 40 different serotypes are described [159].

Rudenko et al. provided data on glycoconjugates using extracted K9 CPS fragments. They report on a substantial cross-reactive antibody response, including protective effectiveness demonstrated by a challenge of immunized mice with a lethal dose of *A. baumannii* K9 [160]. Li et al. demonstrated that their bioconjugate vaccine candidate elicited an efficient immune response and provided protection against an *A. baumannii* infection in a murine sepsis model [161]. Wei et al. employed a synthetically produced pseudaminic acid (Figure 4) conjugated to CRM197. Antibodies raised to their vaccine candidate were able to recognize pseudaminic acid [144]. Among the different research groups, the three different conjugation strategies all provided the ability to come up with effective vaccine designs targeting *A. baumanii* yielding multiple vaccine candidates that show great potential (Table 11).

### 3.5. Pseudomonas aeruginosa

*P. aeruginosa* is a Gram-negative bacterium and primarily opportunistic nosocomial pathogen. It is rated as one of the most common causes of pneumonia in immunocompromised patients, including those with lung disease. Typically cystic fibrosis, HIV-1 and cancer patients often suffer from opportunistic *P. aeruginosa* infections in conjunction with chronic lung infections [163]. Between 2% and 3% of *P. aeruginosa* strains carry the carbapenemase enzyme making them carbapenem-resistant. This resistance particularly increases the risk of mortality among patients with bloodstream infections and leaves very few treatment options [136,137]. *Pseudomonas aeruginosa* carries O-polysaccharide, part of the LPS, as an important variable region and virulence factor. Further characterization of serotyping can be done by applying the International Antigenic Scheme. *P. aeruginosa* can be subdivided into 20 standard O serotypes (O1–O20) based on the structure of O-polysaccharide [164]. Serotypes O5, O6 and O11 are most prevalent in burn wound infections [165], whereas serotypes O6 and O11 are most common in pneumonia [166].

Jamshidi et al. reported on the successful development of a pentasaccharide conjugate from *P. aeruginosa’s* A-Band polysaccharide (Figure 4). Evaluation of the immunogenicity showed that antibodies were able to recognize *P. aeruginosa* LPS [145]. Hegerle et al. investigated a dual-use conjugate vaccine targeting both *P. aeruginosa* (rFlaA and rFlaB, carrier protein) and *K. pneumoniae* (O-polysaccharide O1, O2, O3 and O5, antigens). A quadrivalent conjugate vaccine formulation for the four antigens provided passive protection in both rabbits and mice when challenged [157]. Taken together, these vaccine candidates provide promising preclinical results and provide proof-of-concept for *P. aeruginosa* vaccines (Table 12).

### 3.6. Escherichia coli

Part of the group of *Enterobacter* spp, *E. coli* is a Gram-negative commensal bacterium. It is associated with increasing reports of resistance in most countries and high mortality rates. Resistance has been observed towards third-generation cephalosporins through ESBL or Amp-C beta-lactamase production. *E. coli* is widespread and responsible for community-acquired and hospital-acquired urinary tract infections and bloodstream infections, ventilator-associated pneumonia infections and diarrheal disease as well. Resistance leaves very limited treatment options [136,137]. Serotyping of *E. coli* is based on the capsular polysaccharides, part of the protective structure on their surface. There is significant diversity where more than 180 different O-antigens and approximately 80 K-antigens have been reported [168].

Both Nicolardi et al. and Kowarik et al. (Figure 4) successfully constructed their bioconjugate via the traditional pathways and provide extensive data on the characterization. However, no immunogenicity data was provided making it impossible to assess the effectiveness [72,146]. Both Stevenson et al. and Gening et al. have provided successful immunogenicity data which was discussed for *S. aureus* already regarding the PNAG antigen and which could be applied towards *E. coli* as well [79,151] (Table 13).

## 4. Discussion

Recent developments regarding six selected carriers and their applications are described in the field of conjugate vaccines. OMVs provide a very diverse and flexible platform for the construction of new conjugate vaccines (Table 1). For OMV specifically, we find that there are only a few pathogens used in current approaches for the extraction of OMVs; *Salmonella*, *Shigella* and *Neisseria*. For the future, it is important, with respect to CIES, to include the plethora of other pathogens from which OMVs can be produced [62]: e.g., *A. baumannii* [170], *Bordetella pertussis* [171], *Borrelia burgdorferi* [172], *H. pylori* [173], *K. pneumoniae* [174] and *P. aeruginosa* [175]. Glycoengineered OMVs and proteins go beyond the traditionally extracted OMVs as a carrier. The in vivo conjugation of specific antigens elaborates more on combinations of different pathogens and targets for vaccination (Table 2). The well-known PGCT and OTase-combined expression systems have great potential for new conjugate vaccine development. Whereas less unit operations are required for processing and conjugation of carrier and antigen, heterogeneity of these bioconjugates remains a challenge [72,73]. One direct solution for preventing such heterogeneity would be the more elaborate production of conjugates from individual carrier proteins and synthetically produced oligosaccharides (Table 3). In the field, a lot of progress is made towards synthetically designed oligosaccharides as a replacement for extracted oligosaccharides, aiding in better-controlled production processes [176]. The trade-off between bioconjugation and synthetic oligosaccharide conjugation lies between bioconjugation-related heterogeneity and sometimes longer development times for synthetic oligosaccharides. Other nanoparticles, such as VLPs and protein nanocages, are also studied in the field (Table 4 and Table 5). VLPs can be decorated with different antigens in the form of peptides, proteins, or CPS either through protein display or chemical conjugation. There is a clear advantage for VLPs with the absence of infectivity compared to traditional inactivated viruses and associated risks. Both the full potential of VLPs and protein nanocages as carriers are not yet obvious, with limited variation in the carrier–antigen combinations actively pursued in the field at this time. Synthetically manufactured peptides are finding their application in the cancer field, but also for infectious diseases (Table 6). The main advantage of peptides is that they can be easily designed and manufactured at a large scale and at very high purities with conjugation-ready reactive groups. Additionally, they can be employed both as a carrier and an antigen, used as an adjuvant and provide a specific B- or T-cell-mediated immune response. Peptides might be considered one of the highest potentials in fast vaccine design which could aid in new developments.

With the exciting developments in new carrier proteins, we looked further into the field of vaccination and current and future challenges. With increasing multidrug-resistant strains in the field of AMR, it cannot be overstated that this can be classified as the new (silent) pandemic [134,135]. With the WHO- and CDC-listed ESKAPE pathogens (Table 7), one would urge the development of new vaccines for each of these specific bacteria. Concentrating on recent developments it shows that more examples of the design of synthetic oligosaccharides against the ESKAPE pathogens are found than conjugate vaccines as a whole. Nevertheless, there are, however limited, multiple examples for each of the six pathogens (Table 8, Table 9, Table 10, Table 11, Table 12 and Table 13). While other new carrier proteins are described in this review, their application in the field of ESKAPE pathogens is not obvious. The bioconjugates excluded, it is striking to see that most of the carriers still consist of CRM197 and also TT, which will not be beneficial towards CIES and the effectiveness of newly designed vaccines. When looking at those examples provided, it can be concluded that there are multiple promising vaccine candidates that show favorable preclinical results and provide proof-of-concept for all six ESKAPE pathogens.

## 5. Conclusions

A large influx of new vaccines has been achieved through the use of conjugation strategies. Many conjugate vaccines are developed to prevent infections and diseases. Because of carrier-induced epitope suppression and many AMR targets, traditional carrier proteins, such as TT, DT, CRM197, PD and OMPC, cannot always be used [177]. New carriers will help the development of conjugate vaccines. These carriers can be derived from many different organisms or be produced synthetically and contribute to increased protection against diseases. These carriers will have to be evaluated extensively by physicochemical and immunological methods in combination with the antigen. Looking at recent developments and the data provided in this review, we conclude that all six types of carriers have promising potential. With respect to conjugate vaccines, it was not surprising to find a wide array of new carriers and vaccine candidates to battle different diseases including ESKAPE pathogens. With the lessons learned from COVID-19 and the introduction of mRNA vaccines, we would expect that both vaccine platforms (conjugate and mRNA vaccines) co-exist with conjugate vaccines in the developments to battle ESKAPE pathogens. New vaccine candidates are needed soon to combat infectious diseases.

## 6. Perspective

Conjugate vaccines have proven their application in preventing disease and providing protection through eliciting functional antibodies against infectious diseases. The introduction of new carriers is crucial to provide ample versatility and choice to target particular diseases and AMR targets specifically. It can be expected that the new carriers currently under investigation find their way to clinical trials. Reliable and large-scale production processes of these carriers will be developed within the coming years.

Improvements in the availability and quality of existing polysaccharide antigens for vaccination will largely depend on two platforms. Both bioconjugation and synthetically manufactured polysaccharides show the most potential. Bioconjugation and synthetic routes have their downsides but both platforms eliminate the need for tedious extraction and purification of polysaccharides. As such more reliable and consistently produced bioconjugates will become available. Additionally, new routes for the large-scale production of synthetic oligosaccharides will become available but will have to be financially evaluated in terms of cost-effectiveness. Recent advances in carbohydrate-based antigens and new carriers are currently available. They will enable the swift design of new glycoconjugates targeting AMR and other vaccine-preventable diseases. Lastly, we expect the development of conjugate vaccines in the fight against cancer and neurodegenerative diseases.

## Figures and Tables

**Figure 1 vaccines-11-00219-f001:**
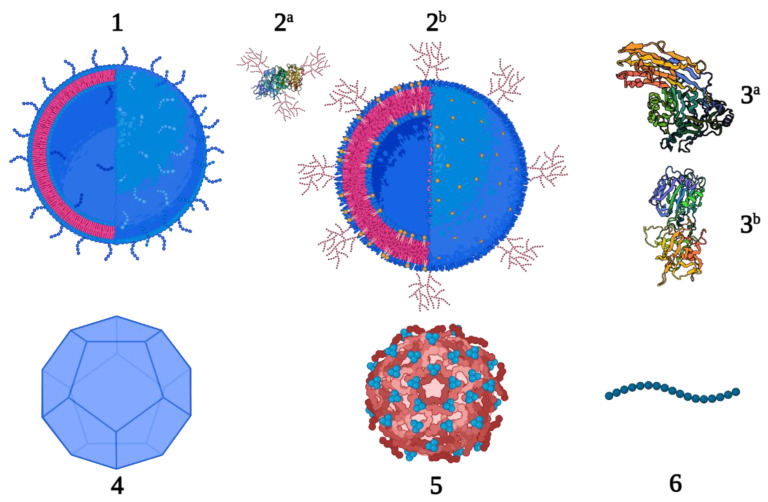
New carrier proteins for conjugate vaccines; OMV and GMMA (**1**), glycoengineered proteins (**2^a^**
*P. aeruginosa* exotoxin A, PDB 1IKQ) and OMV (**2^b^**), proteins (**3^a^**
*Streptococcus pyogenes* streptolysin O, PDB 4HSC and **3^b^** receptor-binding fragment HC of tetanus neurotoxin, PDB 1AF9), VLP (**4**), protein nanocages (**5**) and peptides (**6**).

**Figure 2 vaccines-11-00219-f002:**
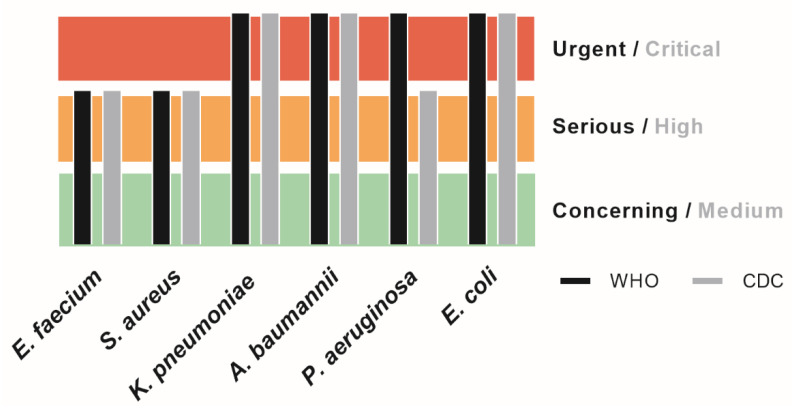
Priority list scaling for the ESKAPE pathogens according to both CDC and WHO.

**Figure 3 vaccines-11-00219-f003:**
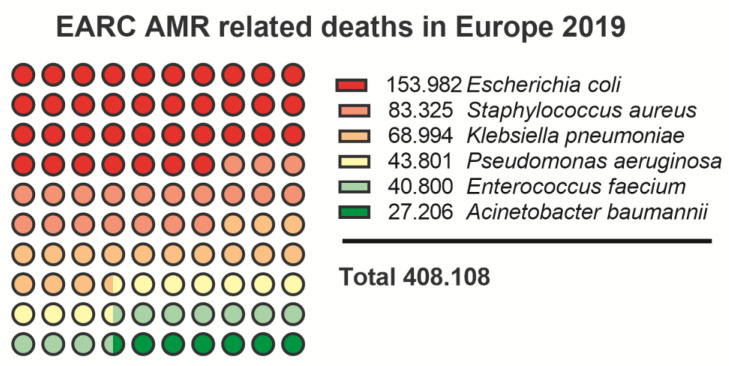
EARC nr. of deaths caused by ESKAPE pathogens in 2019 [138].

**Figure 4 vaccines-11-00219-f004:**
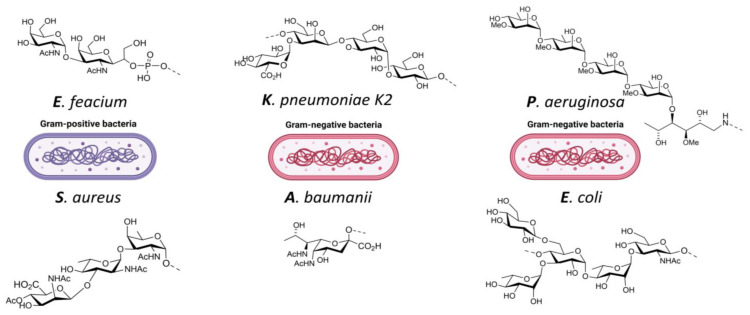
ESKAPE pathogens and examples of target antigens: *E. feacium*: cell wall teichoic acid [141]; *S. aureus*: trisaccharide type 8 capsular polysaccharide [142]; *K. pneumoniae*: capsular polysaccharide K2 [143]; *A. baumanii*: pseudaminic-acid [144]; *P. aeruginosa*: methyl rhamnan pentasaccharide [145]; *E. coli*: serotype O25B [146].

**Table 1 vaccines-11-00219-t001:** OMV/GMMA conjugation strategies.

Vaccine Type	Carrier	Antigen	Chemistry	References
OMV	*S. typhimurium*	SARS-CoV-2 spike receptor-binding domain (RBD)	SpyCatcher-SpyTag ^[1]^	Jiang et al., 2022 [66]
GMMA	*S. typhimurium*	Group A *Streptococcus* cell wall carbohydrate	Reductive amination	Palmieri et al., 2022 [65]
GMMA	*S. typhimurium*	Malaria transmission-blocking protein Pfs25	Oxidation/reductive amination	Di Benedetto et al., 2021 [67]
GMMA	*S. typhimurium*	*P. falciparum* circumsporozoite protein	Thiol-maleimide	Micoli et al., 2020 [53]
	*S. sonnei*	*E. coli* SslE	Reductive amination	
	*N. meningitidis* type B/*S. typhimurium*	*N. meningitidis* serogroups A and C oligosaccharides	SIDEA	
OMV	*N. meningitidis* type B	Malaria transmission-blocking antigen Pfs230	Thiol-maleimide	Scaria et al., 2019 [68]

^[1]^ See protein nanocages for a detailed description of the SpyCatcher-SpyTag ligation.

**Table 2 vaccines-11-00219-t002:** Glycoengineered conjugation strategies.

Vaccine Type	Carrier	Antigen	Chemistry	References
Protein/Bi-valent	*P. aeruginosa* exotoxin A	*E. coli* O2, O6A and O25B	Bioconjugation	Nicolardi et al. [72]
Polysaccharide/Quadrivalent	*P. aeruginosa* exotoxin A	*S. dysenteriae* type O1, O antigen from *S. flexneri* 2a, 3a and 6		Martin et al., 2022 [75]
Polysaccharide/monovalent	*P. aeruginosa* exotoxin A	*S. flexneri 2a* O-polysaccharide		Ravenscroft et al., 2019 [76]
Polysaccharide/protein Bi-valent	*P. aeruginosa* exotoxin A	*K. pneumoniae* K1 and K2 CPSs		Feldman et al., 2019 [77]
Oligosaccharide/mono-, bi-, and trivalent	*E. coli* Acceptor protein ComP	*S. pneumoniae* CPS		Harding et al., 2019 [74]
Polysaccharide/monovalent	*S. pneumoniae* NanA, PiuA, and Sp0148	*S. pneumoniae* serotype 4 CPS		Reglinksy et al., 2018 [78]
OMV	*E. coli* OMV	Poly-N-acetyl-d-glucosamine (rPNAG)		Stevenson et al., 2018 [79]
Polysaccharide/monovalent	*S. paratyphi* A antigenic peptide (P2)	*S. enterica* serovar Paratyphi A O-polysaccharide		Sun et al., 2018 [80]
Polysaccharide/monovalent	*P. aeruginosa* exotoxin A	*F. tularensis* O-antigen		Marshall et al., 2018 [81]

**Table 3 vaccines-11-00219-t003:** Protein-based conjugation strategies.

Vaccine Type	Carrier	Antigen	Chemistry	References
Bivalent	Genetically detoxified Tetanus toxin (8MTT)	Cattle tick fever peptide P0, *H. influenzae* type b (PRP)	Thiol-maleimide, CDAP	Chang et al., 2022 [86]
Bivalent	*S. pneumoniae* serotype type 14 CPS	recombinant SARS-CoV-2 RBD	Reductive amination	Deng et al., 2022 [87]
Monovalent	Group A *Streptococcus* Streptolysin O	Group A *Streptococcus* cell-wall oligosaccharides	Click chemistry	Kapoor et al., 2022 [84]
Bivalent	Rotavirus recombinant ΔVP8 protein	*S*. Typhi capsular polysaccharide (Vi)	EDAC-ADH	Park et al., 2021 [88]
Bivalent	*Streptococcus* C5a peptidase ScpA193, Fn and Fn2	Group A *Streptococcus* cell-wall trisaccharide	di(N-succinimidyl) glutarate	Wang et al., 2021 [83]
Monovalent	*S. aureus* fusion protein (Hla-MntC-ACOL0723)	*S. aureus* 5 (CP5, Reynolds strain) and 8 (CP8, Becker strain)	Carbodiimide	Ahmadi et al., 2020 [89]
Monovalent	*S. typhimurium* flagellin	*S. typhimurium* lipid-A free lipopolysaccharide	Decarboxylative amidation	Chiu et al., 2020 [90]
Monovalent	Recombinant tetanus toxoid heavy chain fragment	HIV-1-fusion peptide (FP8)	Sulfo-SIAB	Ou et al., 2020 [91]
Bivalent	Hepatitis B virus surface antigen	*Pneumococcal* type 33 F-capsular polysaccharide	Carbodiimide	Qian et al., 2020 [92]
Bivalent	Recombinant Tetanus Toxoid Heavy Chain Fragment C	HIV-1 fusion peptide (FP) with eight amino acid residues (FP8)	Amine-to-sulfhydryl, Thiol-maleimide	Yang et al., 2020 [93]
Monovalent	*Enterococcus* secreted antigen A and the peptidyl-prolyl cis-trans isomerase	*E. faecalis* polysaccharide di-heteroglycan	CDAP	Romero-Saavedra et al., 2019 [85]
Bivalent	*Plasmodium falciparum* Pfs25	*S*. Typhi Vi capsular polysaccharide	Carbodiimide	An et al., 2018 [94]
Monovalent	*S. enteritidis* homologous serovar phase 1 flagellin protein	*S. enteritidis* core and O-polysaccharide (COPS)	CDAP	Baliban et al., 2018 [95]
Monovalent	Recombinant Tetanus Toxoid Heavy Chain Fragment C	Oxycodone-based hapten	Carbodiimide	Baruffaldi et al., 2018 [96]
Monovalent, bivalent, and trivalent	ETEC adhesins CFA/I and CS6	*C. jejuni* and *Shigella* polysaccharides and *Shigella flexneri* LPS	TEMPO-mediated oxidation and carbodiimide	Laird et al., 2018 [97]

**Table 4 vaccines-11-00219-t004:** VLP conjugation strategies.

Vaccine Type	Carrier	Antigen	Chemistry	References
Protein display	Qβ	Dengue virus synthetic peptides	Protein display	Warner et al., 2021 [101]
Protein display	Cucumber mosaic virus	Recombinantly expressed SARS-CoV-2 receptor-binding domain	Protein display	Zha et al., 2021 [102]
Protein display	Cucumber mosaic virus	Zika virus E-DIII protein	Protein display	Cabral-Miranda et al., 2019 [103]
Protein display	Qβ	Zika virus MS2 and PP7/Chikungunya virus B-cell synthetic peptides	Protein display	Basu et al., 2018, 2020 [99,100]
Chemical conjugation	Qβ	Thomsen-nouveau antigen, GD2 protein, SARS-CoV-2 peptides	diNHS ester adipate	Sungsuwan et al., 2022 [104]
Chemical conjugation	Recombinant adenoviral type 3 dodecahedron	*S. pneumoniae* serotype 14 CPS trisaccharide	Glutaraldehyde	Prasanna et al., 2021 [105]
Chemical conjugation	Qβ	HIV-1 V1V2 glycopeptide	Click chemistry	Zong et al., 2021 [106]
Chemical conjugation	Qβ	Synthetic Pertussis LPS-like pentasaccharide	diNHS ester adipate	Wang et al., 2020 [107]
Chemical conjugation	Full-length hepatitis B core antigen virus-like particles	Meningococcal group C polysaccharides	Amine-PEG-maleimide	Xu et al., 2019 [108]

**Table 5 vaccines-11-00219-t005:** Protein nanocages conjugation strategies.

Vaccine Type	Carrier	Antigen	Chemistry	References
Tri-valent	*E. coli* Sd-ferritin	SARS-CoV-2 RBD B.1.617.2, D614G and B.1.351	SpyCatcher-SpyTag	Chen et al., 2022 [112]
Monovalent	Horse spleen apoFerritin	Influenza virus PR8 H1N1 hemagglutinin and M2e peptide	Thiol-maleimide	Sheng et al., 2022 [116]
Monovalent	*E. coli* SpyTag–ferritin	ΔN SpyCatcher-fused preS1	SpyCatcher-SpyTag	Wang et al., 2020 [113]
Bivalent	Horse spleen apoFerritin	Influenza virus hemagglutinin	Thiol-maleimide	Wei et al., 2020 [115]

**Table 6 vaccines-11-00219-t006:** Peptide conjugation strategies.

Vaccine Type	Carrier	Antigen	Chemistry	References
Semi-synthetic	IC28 peptide from bacterial flagellin	Recombinant SARS-CoV-2 RBD	Thiol-maleimide	He et al., 2022 [120]
Synthetic	PADRE and polyleucine	*S. pyogenes* (GAS) M-protein-derived B-cell epitopes J8, PL1, and 88/30	Click-chemistry	Azuar et al., 2021 [121]
Synthetic	T-helper cell epitope P25 and polyleucine	Hookworm APR-1 B-cell epitope (p3)	Click-chemistry	Shalash et al., 2021 [122]
Semi-synthetic	S. Typhi OmpC synthetic peptide	*S*. Typhi Vi polysaccharide	ADH-EDC	Haque et al., 2019 [123]

**Table 7 vaccines-11-00219-t007:** AMR for each of the ESKAPE pathogens.

Species	AMR
*Enterococcus faecium*	Vancomycin
*Staphylococcus aureus*	Methicillin and Vancomycin
*Klebsiella pneumoniae*	Carbapenem, ESBL ^[a]^
*Acinetobacter baumannii*	Carbapenem
*Pseudomonas aeruginosa*	Carbapenem
*Escherichia coli*	Carbapenem, ESBL ^[a]^

^[a]^ Extended-spectrum beta-lactamases.

**Table 8 vaccines-11-00219-t008:** Carbohydrate conjugate vaccines targeting *Enterococcus faecium*.

Vaccine Type	Carrier	Antigen	Chemistry	References
Traditional extracted	*Enterococcus* secreted antigen A	Di-heteroglycan	CDAP	Kalfopoulou et al., 2019 [140]
Traditional extracted	*Enterococcus* secreted antigen A and peptidyl-prolyl cis-trans isomerase	Di-heteroglycan	CDAP	Romero-Saavedra et al., 2019 [85]
Semi-synthetic	KLH and HSA	Cell wall teichoic acid	Disuccinimidyl glutarate	Zhou et al., 2017 [141]

**Table 9 vaccines-11-00219-t009:** Carbohydrate conjugate vaccines targeting *Staphylococcus aureus*.

Vaccine Type	Carrier	Antigen	Chemistry	References
Semi-synthetic	Tetanus toxoid, Shiga toxin 1b subunit (Stx1b) and *S. aureus* alpha-hemolysin (Hla H35L)	Synthetic penta- and nona-β-(1→6)-d-glucosamine (PNAG)	Thiol-maleimide	Gening et al., 2021 [151]
Semi-synthetic	CRM197	Capsular polysaccharide type 8 trisaccharide	Bis(p-nitrophenyl adipate)	Zhao et al., 2020 [142]
Bioconjugate	*E. coli* OMV	Poly-N-acetyl-d-glucosamine (rPNAG)	Bioconjugation	Stevenson et al., 2018 [79]

**Table 10 vaccines-11-00219-t010:** Carbohydrate conjugate vaccines targeting *Klebsiella pneumoniae*.

Vaccine Type	Carrier	Antigen	Chemistry	References
Traditional extracted	Polylactic-co-glycolic acid (PLGA)	*K. pneumoniae* K2O1 capsule antigen	W/O/W emulsion	Ghaderinia et al., 2022 [155]
Traditional extracted	CRM197	Capsular polysaccharide K1 and K2	Thiol-maleimide	Lin et al., 2022 [143]
Bioconjugate	*E. coli*	*K. pneumoniae* serotype O2 polysaccharide	Bioconjugation	Peng et al., 2021 [156]
Semi-synthetic	CRM197	K2 hexa-, hepta-, and octa-saccharide	Thiol-maleimide	Ravinder et al., 2020 [154]
Bioconjugate	*P. aeruginosa* exotoxin A	Capsular polysaccharide K1 and K2	Bioconjugation	Feldman et al., 2019 [77]
Traditional extracted	*P. aeruginosa* rFlaA and rFlaB	O-polysaccharide O1, O2, O3 and O5	Thiol-maleimide	Hegerle et al., 2018 [157]
Semi-synthetic	CRM197	Synthetic hexasasaccharide	p-nitrophenyl adipate ester	Seeberger et al., 2017 [158]

**Table 11 vaccines-11-00219-t011:** Carbohydrate conjugate vaccines targeting *Acinetobacter baumanii*.

Vaccine Type	Carrier	Antigen	Chemistry	References
Traditional extracted	BSA, OVA, KLH	CPS K9 di- and trimers	Reductive amination, squaric acid	Rudenko et al., 2022 [160]
Bioconjugate	cholera toxin B subunit	O-linked PgIS	Bioconjugation	Li et al., 2022 [161]
Semi-synthetic	CRM197	Pseudaminic acid	OPA (ortho-phthalaldehyde)	Wei et al., 2021 [144]
Semi-synthetic	CRM197	Pseudaminic acid	Thiol-maleimide	Lee et al., 2018 [162]

**Table 12 vaccines-11-00219-t012:** Carbohydrate conjugate vaccines targeting *Pseudomonas aeruginosa*.

Vaccine Type	Carrier	Antigen	Chemistry	References
Semi-synthetic	PLGA	*P. aeruginosa* LPS and OPS	Carbodiimide	Maleki et al., 2022 [167]
Semi-synthetic	HSA	Methyl Rhamnan Pentasaccharide	Reductive amination	Jamshidi et al., 2022 [145]
Traditional extracted	*P. aeruginosa* rFlaA and rFlaB	O-polysaccharide O1, O2, O3 and O5	Thiol-maleimide	Hegerle et al., 2018 [157]

**Table 13 vaccines-11-00219-t013:** Carbohydrate conjugate vaccines targeting *Escherichia coli*.

Vaccine Type	Carrier	Antigen	Chemistry	References
Semi-synthetic	Tetanus toxoid, Shiga toxin 1b subunit (Stx1b) and *S. aureus* alpha-hemolysin (Hla H35L)	Synthetic penta- and nona-β-(1→6)-d-glucosamine (PNAG)	Thiol-maleimide	Gening et al., 2021 [151]
Bioconjugate	*P. aeruginosa* Exotoxin A	*E. coli* O25B	Bioconjugation	Kowarik et al., 2021 [146]
Bioconjugate	*P. aeruginosa* exotoxin A	O1, O2, O4, O6, O8, O15, O16, O18, O25 and O75	Bioconjugation	Saade et al., 2020 [169]
Bioconjugate	*P. aeruginosa* Exotoxin A	*E. coli* O2, O6A and O25B	Bioconjugation	Nicolardi et al. [72]
Bioconjugate	*E. coli* OMV	Poly-N-acetyl-d-glucosamine (rPNAG)	Bioconjugation	Stevenson et al., 2018 [79]
Semi-synthetic	Tetanus toxoid, Shiga toxin 1b subunit (Stx1b) and *S. aureus* alpha-hemolysin (Hla H35L)	Synthetic penta- and nona-β-(1→6)-D-glucosamine (PNAG)	Thiol-maleimide	Gening et al., 2021 [151]

## Data Availability

Not applicable.

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
