# Peer review of "Carriers and Antigens: New Developments in Glycoconjugate Vaccines"

_vaccines, 2023, doi:10.3390/vaccines11020219_

Round 1

Reviewer 1 Report

In the review article entitled "Carriers and antigens: new developments in glycoconjugate vaccines" by van der Put et al., the authors present a comprehensive compilation of the state of the art regarding glycoconjugate vaccines. Firstly, the work covers the different carriers employed in this kind of vaccines, with an emphasis on the more recent developments. Then, they focus the attention on the counterpart of conjugated vaccines: the glyco-antigens, with an update on the most serious MDR ESKAPE pathogens, nicely illustrated by figure 4. Overall, the manuscript contains four figures, 13 tables and 176 references that are relevant to this topic and complement the main text, a sum that I consider appropriate for the length of this work. This review is well written and well structured, and I found it very informative, not only for researchers directly working on this topic, but also for a more general audience working in the chemical biology field. For these reasons I would like to commend the authors on a fine job and to recommend it be published as is.

Author Response

Dear Editor, Reviewer,

We are happy to learn that no further action is required with the comment to publish as is.

Reviewer 2 Report

In Introduction describe and reference the first work on immunogenicity of polysaccharide conjugates by Avery, Heidelberger (beginning of 1900s): https://pubmed.ncbi.nlm.nih.gov/?term=Avery+OT+Heidelberger+M&sort=date

Line 125: Clostridium difficile is now Clostridioides difficile

When describing bioconjugation, it would be helpful to mention that bioconjugation is only possible after the structure of the polysaccharide has been identified, and subsequently the gene cluster responsible for its production, and that in certain bacteria (such as Helicobacter pylori) there are no gene clusters responsible for the biosynthesis of the polysaccharide (genes are widely distributed in the genome).

Author Response

Dear Editor, Reviewer,

We welcome the provided feedback from all reviewers. We have made the necessary improvements to our manuscript. We have made the following changes to the manuscript which we describe below;

Reviewer 2

Point 1:

Q: In Introduction describe and reference the first work on immunogenicity of polysaccharide conjugates by Avery, Heidelberger (beginning of 1900s): https://pubmed.ncbi.nlm.nih.gov/?term=Avery+OT+Heidelberger+M&sort=date

A: We have added the reference for Avery and Goebel (line 38), which describes the immunological evaluation of conjugates.

Point 2:

Q: Line 125: Clostridium difficile is now Clostridioides difficile

A: We have changed Clostridium difficile to Clostridioides difficile.

Point 3:

Q: When describing bioconjugation, it would be helpful to mention that bioconjugation is only possible after the structure of the polysaccharide has been identified, and subsequently the gene cluster responsible for its production, and that in certain bacteria (such as Helicobacter pylori) there are no gene clusters responsible for the biosynthesis of the polysaccharide (genes are widely distributed in the genome).

A: At §2.3.2 at line 180 and 181 we have added the suggested improvements for bioconjugation that the structure of the polysaccharide has to be identified including the gene cluster for its production.

Reviewer 3 Report

In this review the authors have described the advances of glycoconjugate vaccines in the last five years focusing on the developments of both new carriers and antigens. 

This review has been well designed and written. It is clear and fixes all the critical points.

Major comments:

1. No information has been reported about the MAPS technology (Zhang F, Lu YJ, Malley R. Proc Natl Acad Sci U S A. 2013 Aug 13;110(33):13564-9.) Could the authors describe this technology? One 24-valent vaccine is currently in clinical trial.

2. The critical issue revealed for bioconjugate vaccines (i.e. limited number of saccharide chains per mol of protein) has not been clearly described. Could the authors integrate this section?

3. The industrial feasibility for synthetic oligosaccharide but be a critical point for final acceptance. Could the authors expand this section?

Author Response

Dear Editor, Reviewer,

We welcome the provided feedback from all reviewers. We have made the necessary improvements to our manuscript. We have made the following changes to the manuscript which we describe below;

Reviewer 3

Point 1:

Q: No information has been reported about the MAPS technology (Zhang F, Lu YJ, Malley R. Proc Natl Acad Sci U S A. 2013 Aug 13;110(33):13564-9.) Could the authors describe this technology? One 24-valent vaccine is currently in clinical trial.

A: We appreciate the suggestion for this interesting conjugation strategy. This review is focusing on carriers and antigens, and not conjugation chemistry persé. Additionally, inclusion criteria comprised of more recent research (2018-2022), for this we believe the mentioned reference from 2013 is out of scope.

Point 2:

Q: The critical issue revealed for bioconjugate vaccines (i.e. limited number of saccharide chains per mol of protein) has not been clearly described. Could the authors integrate this section?

A: We have evaluated § 2.3.2 where we elaborate on bioconjugation. We have added "themselves" to line 188 to put focus on the polysaccharide heterogeneity. At line 190 we added "single" to put focus on the conjugation rate per protein.

Point 3:

Q:  The industrial feasibility for synthetic oligosaccharide but be a critical point for final acceptance. Could the authors expand this section?

A: While the financial evaluation and industrial feasibility are not part of this review, we have added a statement at the perspectives section at line 585 - 588, that these will have to be evaluated.